# A Probabilistic Approach to Constrained Deep Clustering.

## Abstract

Clustering with constraints has gained significant attention in the field of semi-supervised machine learning as it can leverage partial prior information on a growing amount of unlabelled data. Following recent advances in deep generative models, we derive a novel probabilistic approach to constrained clustering that can be trained efficiently in the framework of stochastic gradient variational Bayes. In contrast to existing approaches, our model (CVaDE) uncovers the underlying distribution of the data conditioned on prior clustering preferences, expressed as *pairwise constraints*. The inclusion of such constraints allows the user to guide the clustering process towards a desirable partition of the data by indicating which samples should or should not belong to the same class. We provide extensive experiments to demonstrate that CVaDE shows superior clustering performances and robustness compared to state-of-the-art deep constrained clustering methods in a variety of data sets. We further demonstrate the usefulness of our approach on challenging real-world medical applications and face image generation.

## 1 Introduction

The ever-growing amount of data and the time cost associated with its labeling has made clustering a relevant task in the field of machine learning. Yet, in many cases, a fully unsupervised clustering algorithm might naturally find a solution which is not consistent with the domain knowledge (Basu et al., 2008). In medicine, for example, clustering could be driven by unwanted bias, such as the type of machine used to record the data, rather than more informative features. Moreover, practitioners often have access to prior information about the types of clusters that are sought, and a principled method to guide the algorithm towards a desirable configuration is then needed. *Constrained clustering*, therefore has a long history in machine learning as it enforces desirable clustering properties by incorporating domain knowledge, in the form of constraints, into the clustering objective.

Following recent advances in deep clustering, constrained clustering algorithms have been recently used in combination with deep neural networks (DNN) to favor a better representation of high-dimensional data sets. The methods proposed so far mainly extend some of the most widely used deep clustering algorithms, such as DEC (Xie et al., 2016), to include a variety of loss functions that force the clustering process to be consistent with the given constraints (Ren et al., 2019; Shukla et al., 2018; Zhang et al., 2019b). Although they perform well, none of the above methods model the data generative process. As a result, they can neither uncover the underlying structure of the data, nor control the strength of the clustering preferences, nor generate new samples (Min et al., 2018).

To address the above issues, we propose a novel probabilistic approach to constrained clustering, the Constrained Variational Deep Embedding (CVaDE), that uncovers the underlying data distribution conditioned on domain knowledge, expressed in the form of pairwise constraints. Our method extends previous work in unsupervised variational deep clustering (Jiang et al., 2017; Dilokthanakul et al., 2016) to incorporate clustering preferences as Bayesian prior probabilities with varying degrees of uncertainty. This allows systematical reasoning about parameter uncertainty (Zhang et al., 2019a), thereby enabling the ability to perform Bayesian model validation, outlier detection and data generation. By integrating prior information in the generative process of the data, our model can guide the clustering process towards the configuration sought by the practitioners.

**Our main contributions** are as follows: (i) We propose a constrained clustering method (CVaDE) to incorporate given clustering preferences, with varying degrees of certainty, within the Variational

Auto-Encoder (VAE) framework. (ii) We provide a thorough empirical assessment of our model. In particular, we show that (a) a small fraction of prior information remarkably increases the performance of CVaDE compared to unsupervised variational clustering methods, (b) our model shows superior clustering performance compared to state-of-the-art deep constrained clustering models on a wide range of data sets and, (c) our model proves to be robust against noise as it can easily incorporate the uncertainty of the given constraints. (iii) We show that our model can drive the clustering performance towards different desirable configurations, depending on the constraints used, and it successfully generates new samples on challenging real-world image data.

## 2 THEORETICAL BACKGROUND & RELATED WORK

**Constrained Clustering.** A constrained clustering problem differs from the classical clustering scenario as the user has access to some pre-existing knowledge about the desired partition of the data. The constraints are usually expressed as pairwise constraints (Wagstaff & Cardie, 2000), consisting of *must-links* and *cannot-links*, which indicate whether two samples are believed to belong to the same cluster or to different clusters. Such pairwise relations contain less information than the labels used in classification tasks but are usually easier to obtain. Traditional clustering methods have been then extended to enforce pairwise constraints (Lange et al., 2005). COP-KMEANS (Wagstaff et al., 2001) and MPCK-mean (Bilenko et al., 2004) adapted the well-known K-means algorithm, while several methods proposed a constrained version of the Gaussian Mixture Models (Shental et al., 2003; Law et al., 2004; 2005). Among them, penalized probabilistic clustering (PPC, Lu & Leen (2004)) is most related to our work as it expresses the pairwise constraints as Bayesian priors over the assignment of data points to clusters. However, PPC, as well as the previous models, shows poor performance and high computational complexity on high-dimensional and large-scale data sets.

**Deep Constrained Clustering.** To overcome the limitations of the above models, constrained clustering algorithms have been lately used in combination with DNNs. Hsu & Kira (2015) train a DNN to minimize the Kullback-Leibler (KL) divergence between similar pairs of samples, while Chen (2015) performs semi-supervised maximum margin clustering of the learned features on a DNN. More recently, many extensions of the widely used DEC model (Xie et al., 2016) have been proposed to include a variety of loss functions to enforce pairwise constraints. Among them, SDEC (Ren et al., 2019) includes a distance loss function that forces the data points with a must-link to be close in the latent space and vice-versa. C-IDEC (Zhang et al., 2019b), uses, instead, a KL divergence loss, extending the work of Shukla et al. (2018). Other works have focused on discriminative clustering methods by self-generating pairwise constraints from either Siamese networks or KNN graphs (Smieja et al., 2020) (Fogel et al., 2019). As none of the approaches proposed so far is based on generative models, the above methods fail to uncover the underlying data distribution. Additionally, DEC-based architectures rely on heavy pretraining of the autoencoder, resulting in no theoretical guarantee that the learned latent space is indeed suitable for clustering (Min et al., 2018).

**VAE-based deep clustering.** Many models have been proposed in the literature to perform unsupervised clustering through deep generative models (Li et al., 2019; Yang et al., 2019; Manduchi et al., 2019; Kopf et al., 2019). Among them, the Variational Deep Embedding (VaDE, Jiang et al. (2017)) and the Gaussian Mixture Variational Autoencoder (GMM-VAE, Dilokthanakul et al. (2016)) propose a variant of the VAE (Kingma & Welling (2014); Rezende et al. (2014)) in which the prior is a Gaussian Mixture distribution. With this assumption, they construct an inference model that can be directly optimised in the framework of stochastic gradient variational Bayes. However, variational deep clustering methods, such as the VaDE, cannot incorporate domain knowledge and clustering preferences. Even though a semi-supervised version on the VAE has been proposed by Kingma et al. (2014), the latter cannot be naturally applied to clustering. For this reason, we aim at extending the above methods to incorporate clustering preferences in the form of constraints, modeled as Bayesian priors, to guide the clustering process towards a desirable configuration.

## 3 CONSTRAINED VARIATIONAL DEEP EMBEDDING

In the following section, we propose a novel constrained clustering model (CVaDE) to incorporate clustering preferences, with varying degree of certainty, in a VAE-based deep clustering setting. In particular, we use the VaDE (Jiang et al., 2017) generative assumptions of the data, conditioned on

the domain knowledge. We then illustrate how our model can be trained efficiently in the framework of stochastic gradient variational Bayes by optimizing the Conditional Variational Lower Bound. Additionally, we define concrete prior formulations to incorporate our preferences, with a focus on pairwise constraints.

### 3.1 THE GENERATIVE ASSUMPTIONS

Let us consider a data set $\boldsymbol{X} = \{\boldsymbol{x}_i\}_{i=1}^N$ consisting of $N$ samples with $\boldsymbol{x}_i \in \mathbb{R}^M$ that we wish to cluster into $K$ groups according to some prior information encoded as $\boldsymbol{G}$. For example, we may know *a priori* that certain samples should be clustered together with different degree of certainty. Hence $\boldsymbol{G}$ encodes both our prior knowledge on the data set and the degree of confidence.

We assume the data is generated from a random process consisting of three steps. First, the cluster assignments $\mathbf{c} = \{c_i\}_{i=1}^N$, with $c_i \in \{1, \ldots, K\}$, are sampled from a distribution conditioned on the prior information, $\mathbf{c} \sim p(\mathbf{c}|\boldsymbol{G})$. Next, for each cluster assignment $c_i$, a continuous latent embedding, $\mathbf{z}_i \in \mathbb{R}^D$, is sampled from a Gaussian distribution, whose mean and variance depend on the selected cluster $c_i$. Finally, the sample $\mathbf{x}_i$ is generated from a distribution conditioned on $\mathbf{z}_i$. Given $c_i$, the generative process can be summarized as:

$$\mathbf{z}_i \sim p(\mathbf{z}_i|c_i) = \mathcal{N}(\mathbf{z}_i|\boldsymbol{\mu}_{c_i}, \boldsymbol{\sigma}_{c_i}^2 \mathbb{I}) \tag{1}$$

$$\mathbf{x}_i \sim p_\theta(\mathbf{x}_i|\mathbf{z}_i) = \begin{cases} \mathcal{N}(\mathbf{x}_i|\boldsymbol{\mu}_{x_i}, \boldsymbol{\sigma}_{x_i}^2 \mathbb{I}) & \text{with} \quad [\boldsymbol{\mu}_{x_i}, \boldsymbol{\sigma}_{x_i}^2] = f(\mathbf{z}_i; \boldsymbol{\theta}) \text{ if } \mathbf{x}_i \text{ is real-valued} \\ Ber(\boldsymbol{\mu}_{x_i}) & \text{with} \quad \boldsymbol{\mu}_{x_i} = f(\mathbf{z}_i; \boldsymbol{\theta}) \text{ if } \mathbf{x}_i \text{ is binary} \end{cases} \tag{2}$$

where $\boldsymbol{\mu}_{c_i}$ and $\boldsymbol{\sigma}_{c_i}^2$ are mean and variance of the Gaussian distribution corresponding to cluster $c_i$ in the latent space and the function $f(\boldsymbol{z}; \boldsymbol{\theta})$ is a neural network, called *decoder*, parametrized by $\boldsymbol{\theta}$.

Without prior information, that is when $p(\mathbf{c}|\boldsymbol{G}) = p(\mathbf{c}) = \prod_i p(c_i) = \prod_i Cat(c_i|\boldsymbol{\pi})$, the cluster assignments are independent and identical distributed as they follow a categorical distribution with mixing parameters $\boldsymbol{\pi}$. In that case, the generative assumptions described above are equal to those of Jiang et al. (2017) and the parameters of the model can be learned using the unsupervised VaDE method (see Appendix C). In the following, we explore the case when $p(\mathbf{c}|\boldsymbol{G}) \neq p(\mathbf{c})$.

### 3.2 CONDITIONAL VARIATIONAL LOWER BOUND

Given the data generative assumptions illustrated in Sec. 3.1, the objective is to infer the parameters $\boldsymbol{\pi}$, $\boldsymbol{\mu}_c$, $\boldsymbol{\sigma}_c^2$ and $\boldsymbol{\theta}$ which better explain the data $\mathbf{X}$ given prior information on the cluster assignments $\boldsymbol{G}$. We achieve this by maximizing the marginal log-likelihood conditioned on $\boldsymbol{G}$, that is:

$$\log p(\mathbf{X}|\boldsymbol{G}) = \log \int_{\mathbf{Z}} \sum_{\mathbf{c}} p(\mathbf{X}, \mathbf{Z}, \mathbf{c}|\boldsymbol{G}), \tag{3}$$

where $\mathbf{Z} = \{\boldsymbol{z}_i\}_{i=1}^N$ is the collection of the latent embeddings corresponding to the data set $\boldsymbol{X}$. The conditional joint probability is derived from Eq 1/2 and can be factorized as:

$$p(\mathbf{X}, \mathbf{Z}, \mathbf{c}|\boldsymbol{G}) = p_\theta(\mathbf{X}|\mathbf{Z})p(\mathbf{Z}|\mathbf{c})p(\mathbf{c}|G) = p(\mathbf{c}|\boldsymbol{G}) \prod_{i=1}^N p_\theta(\mathbf{x}_i|\mathbf{z}_i)p(\mathbf{z}_i|c_i). \tag{4}$$

Since the conditional log-likelihood is intractable, we derive a lower bound of the log marginal conditional probability of the data, which we call Conditional ELBO (C-ELBO, $\mathcal{L}_C$):

$$\mathcal{L}_C(\mathbf{X}|\boldsymbol{G}) = \mathbb{E}_{q_\phi(\mathbf{z}, \mathbf{c}|\mathbf{X})} \log \frac{p(\mathbf{X}, \mathbf{Z}, \mathbf{c}|\boldsymbol{G})}{q_\phi(\mathbf{Z}, \mathbf{c}|\mathbf{X})} \tag{5}$$

Similarly to Jiang et al. (2017) and Dilokthanakul et al. (2016), we employ the following amortized mean-field variational distribution:

$$q_\phi(\mathbf{Z}, \mathbf{c}|\mathbf{X}) = q_\phi(\mathbf{Z}|\mathbf{X})p(\mathbf{c}|\mathbf{Z}) = \prod_{i=1}^N q_\phi(\mathbf{z}_i|\mathbf{x}_i)p(c_i|\mathbf{z}_i) \quad \text{with} \quad p(c_i|\mathbf{z}_i) = \frac{p(\mathbf{z}_i|c_i)p(c_i)}{\sum_k p(\mathbf{z}_i|k)p(k)}, \tag{6}$$

where $q_\phi(\mathbf{z}_i|\mathbf{x}_i)$ is a Gaussian distribution with mean $\mu(\boldsymbol{x}_i)$ and variance $\sigma^2(\boldsymbol{x}_i)\mathbb{I}$ which are the outputs of a neural network, called *encoder*, parametrized by $\phi$ and $p(c_i = k)$ is denoted as $p(k)$ for simplicity. It is important to note that, in this formulation, the variational distribution does not depend on $\mathbf{G}$. This approximation is used to retain a mean-field variational distribution if the cluster assignments, conditioned on the prior information, are not independent (Sec 3.4.1), that is when $p(\mathbf{c}|\boldsymbol{G}) \neq \prod_i p(c_i|\boldsymbol{G})$.

## 3.3 ROLE OF THE PRIOR INFORMATION

To highlight how the prior information $\boldsymbol{G}$ influences the clustering objective, we rewrite Eq. 5 as:

$$\mathcal{L}_{\mathrm{C}}(\mathbf{X}|\boldsymbol{G}) = \mathbb{E}_{q_\phi(\mathbf{Z}|\mathbf{X})}\left[\log p_\theta(\mathbf{X}|\mathbf{Z})\right] - D_{KL}(q_\phi(\mathbf{Z},\mathbf{c}|\mathbf{X})\|p(\mathbf{Z},\mathbf{c}|\boldsymbol{G})). \tag{7}$$

The first term is called the *reconstruction term*, similarly to the VAE. The second term, on the other hand, is the Kullback-Leibler (KL) divergence between the variational posterior and the Constrained Gaussian Mixture prior. By maximizing the C-ELBO, the variational posterior mimics the true conditional probability of the latent embeddings and the cluster assignments. This results in enforcing the latent embeddings to follow a Gaussian mixture which agrees on the clustering preferences.

Using Eq. 6, the Conditional ELBO can be further factorized as:

$$\begin{aligned}\mathcal{L}_{\mathrm{C}}(\mathbf{X}|\boldsymbol{G}) =& E_{p(\mathbf{c}|\mathbf{Z})}[\log p(\mathbf{c}|G)] + E_{q_\phi(\mathbf{Z}|\mathbf{X})}[\log p_\theta(\mathbf{X}|\mathbf{Z})]\\ &+ E_{q_\phi(\mathbf{Z}|\mathbf{X})p(\mathbf{c}|\mathbf{Z})}[\log p(\mathbf{Z}|\mathbf{c})] - E_{q_\phi(\mathbf{Z},\mathbf{c}|\mathbf{X})}[\log q_\phi(\mathbf{Z},\mathbf{c}|\mathbf{X})],\end{aligned} \tag{8}$$

where the last three terms are not affected by $\boldsymbol{G}$ and they can be rewritten using the SGVB estimator and the reparameterization trick (Kingma & Welling, 2014) to be trained efficiently using stochastic gradient descent (see Appendix B). The first term, on the other hand, is investigated in Sec 3.4.

## 3.4 CONDITIONAL PRIOR PROBABILITY

We incorporate our clustering preference through the conditional probability $p(\mathbf{c}|\boldsymbol{G})$. In particular, we construct the conditional prior probability to be:

$$p(\mathbf{c}|\boldsymbol{G}) = \frac{\prod_i \pi_{\mathrm{c}_i} g_i(\mathbf{c})}{\sum_{\mathbf{c}} \prod_j \pi_{\mathrm{c}_j} g_j(\mathbf{c})} = \frac{1}{\Omega(\boldsymbol{\pi})} \prod_i \pi_{\mathrm{c}_i} g_i(\mathbf{c}), \tag{9}$$

where $\boldsymbol{\pi} = \{\pi_k\}_{k=1}^K$ are the weights associated to each cluster, $\mathrm{c}_i$ is the cluster assignment of sample $\mathbf{x}_i$, $\Omega(\boldsymbol{\pi})$ is the normalization factor and $g_i(\boldsymbol{c})$ is a weighting function that assumes large values if $\mathrm{c}_i$ agrees with our belief with respect to $\mathbf{c}$ and low values otherwise.

### 3.4.1 PAIRWISE CONSTRAINTS

We hereby focus on expressing the conditional prior distribution in the context of pairwise constrained clustering and we do so by adapting the work of Lu & Leen (2004) in our variational framework. The weighting function $g_i(\mathbf{c})$ is then defined as:

$$g_i(\mathbf{c}) = \prod_{j\neq i} \exp\left(\boldsymbol{W}_{i,j}\delta_{\mathrm{c}_i \mathrm{c}_j}\right), \tag{10}$$

where $\delta$ is the Kronecker $\delta$-function and $\boldsymbol{W} \in \mathbb{R}^{N\times N}$ is a symmetric matrix containing the pairwise preferences and confidence. In particular, $\boldsymbol{W}_{i,j} = 0$ if we have no prior information on samples $\boldsymbol{x}_i$ and $\boldsymbol{x}_j$, $\boldsymbol{W}_{i,j} > 0$ if there is a *must-link* constraint (the two samples should be clustered together) and $\boldsymbol{W}_{i,j} < 0$ if there is a *cannot-link* constraint (the two samples should not be clustered together). The value $|\boldsymbol{W}_{i,j}| \in [0,\infty)$ reflects the degree of certainty in the constraint. For example, if $\boldsymbol{W}_{i,j} \to -\infty$ then $\boldsymbol{x}_i$ and $\boldsymbol{x}_j$ must be assigned to different clusters otherwise $p(\mathbf{c}|\boldsymbol{G}) \to 0$ (*hard* constraint). On the other hand, smaller values indicate a *soft preference* as they admit some degree of freedom in the model. An heuristic to select $|\boldsymbol{W}_{i,j}|$ is presented in Sec 4. Interestingly, the probability $p(\mathbf{c}|\boldsymbol{G})$ with pairwise constraints can be seen as the posterior of the superparamagnetic clustering method (Blatt et al., 1996), with loss function given by a fully connected Potts model (Wu, 1982). Differently from our method, Blatt et al. (1996) cluster the data according to the pairwise correlation functions $\mathbb{E}_{p(\mathbf{c}|\boldsymbol{G})}\delta_{c_i c_j}$ that are estimated with MCMC methods.

Finally, we incorporate the conditional prior distribution in Eq. 8. The first term can be written as:

$$\mathbb{E}_{p(\mathbf{c}|\mathbf{Z})}\left[\log p(\mathbf{c}|\boldsymbol{G})\right] = \mathbb{E}_{p(\mathbf{c}|\mathbf{Z})} \log \frac{1}{\Omega(\boldsymbol{\pi})} \prod_i \pi_{\mathrm{c}_i} \prod_{j\neq i} \exp\left(\boldsymbol{W}_{i,j}\delta_{\mathrm{c}_i \mathrm{c}_j}\right) \tag{11}$$

$$= -\log\Omega(\boldsymbol{\pi}) + \sum_i \mathbb{E}_{p(\mathrm{c}_i|\mathbf{z}_i)} \log\pi_{\mathrm{c}_i} + \sum_{i,j\neq i} \mathbb{E}_{p(\mathrm{c}_i|\mathbf{z}_i)}\mathbb{E}_{p(\mathrm{c}_j|\mathbf{z}_j)} \boldsymbol{W}_{i,j}\delta_{\mathrm{c}_i \mathrm{c}_j} \tag{12}$$

$$= -\log\Omega(\boldsymbol{\pi}) + \sum_i \sum_k p(\mathrm{c}_i=k|\mathbf{z}_i)\log\pi_{\mathrm{c}_i} + \sum_{i,j\neq i}\sum_k p(\mathrm{c}_i=k|\mathbf{z}_i)p(\mathrm{c}_j=k|\mathbf{z}_j)\boldsymbol{W}_{i,j}. \tag{13}$$

Maximizing Eq. 13 w.r.t. $\boldsymbol{\pi}$ poses computational problems due to the normalization factor $\Omega(\boldsymbol{\pi})$. Crude approximations are investigated in (Basu et al., 2008), however we choose to fix the parameter $\pi_k = 1/K$ to make $\mathbf{z}$ uniformly distributed in the latent space, as in previous works (Dilokthanakul et al., 2016). By doing so, the normalization factor can be treated as a constant. The analysis of different approaches to learn the weights $\boldsymbol{\pi}$ is left for future work. The C-ELBO with a pairwise constrained prior can then be optimized using Monte Carlo sampling and stochastic gradient descent.

### 3.4.2  FURTHER POSSIBLE CONSTRAINTS

Given the flexibility of our general framework, different types of constraints can be included in the formulation of the weighting functions $g_i(\mathbf{c})$. In particular, we can perform semi-supervised learning by setting the prior information as:

$$g_i(\mathbf{c}) = g(\mathbf{c}_i) = \exp\left(\boldsymbol{W}_{i,k}\right) \quad \text{with} \quad \boldsymbol{W} \in \mathbb{R}^{N \times K} \tag{14}$$

where $\boldsymbol{W}_{i,k}$ indicates whether the sample $x_i$ should be assigned to cluster $k$.

Additionally, one could also include triple-constraints by modifying the weighting function to be:

$$g_i(\mathbf{c}) = \prod_{j,k \neq i} \exp\left(\boldsymbol{W}_{i,j,k}\delta_{\mathbf{c}_i\mathbf{c}_j\mathbf{c}_k}\right) \quad \text{with} \quad \boldsymbol{W} \in \mathbb{R}^{N \times N \times N} \text{ symmetric.} \tag{15}$$

where $\boldsymbol{W}_{i,j,k} = 0$ if we do not have any prior information, $\boldsymbol{W}_{i,j,k} > 0$ indicates that the samples $\boldsymbol{x}_i$, $\boldsymbol{x}_j$ and $\boldsymbol{x}_k$ should be clustered together and $\boldsymbol{W}_{i,j,k} < 0$ if they should belong to different clusters. The analysis of these different constraints formulation is outside the scope of our work but they may represent interesting directions for future work.

## 4  EXPERIMENTS

In the following, we provide a thorough empirical assessment of our proposed method (CVaDE) with pairwise constraints using a wide range of data sets. First, we evaluate our model's performance compared to both unsupervised variational deep clustering methods and state-of-the-art constrained clustering methods. Then, we present extensive evidence of the ability of our model to handle noisy constraint information. We additionally perform experiments on a medical application to prove that our model can reach different desirable partitions of the data, depending on the constraints used, even with real-world, noisy data. Finally, we show that CVaDE successfully generates new data, using the learned generative process of Sec 3.1, on a challenging face image data set.

**Baselines & implementation details.** As baselines, we include the traditional constrained K-means (MPCK-means, Bilenko et al. (2004)) and two recent deep constrained clustering methods based on DEC (SDEC, Ren et al. (2019), and C-IDEC, Zhang et al. (2019b)) as they achieve state-of-the-art performance in constrained clustering. We also compare our model to the unsupervised variational deep clustering method VaDE (Jiang et al., 2017). To implement our model, we were careful in maintaining a fair comparison with the baselines. In particular, we adopted the same encoder and decoder feed-forward architecture among all methods with four layers of 500, 500, 2000, $D$ units respectively, where $D = 10$ unless stated otherwise. The VAE is pretrained for 10 epochs while the DEC-based baselines need a more complex layer-wise pretraining of the autoencoder which involves 50 epochs of pretraining for each layer and 100 epochs of pretraining as finetuning. The pairwise constraints are chosen randomly within the training set by sampling two data points and assigning a must-link if they have the same label and a cannot-link otherwise. Unless stated otherwise, the values of $|W_{i,j}|$ are set to $10^4$ for all data sets, and $N$ pairwise constraints are used for both our model and the constrained clustering baselines, where $N$ is the number of samples of the considered data set. Note that $N$ *pairwise constraints* can be obtained by using only $\sqrt{N}$ *labeled* data points. To allow for fast iteration we simplify the last term of Eq. 13 by allowing the search of pairwise constraints to be performed only inside the considered batch. We observed empirically that, with a batch size of 1024, this approximation does not affect the clustering performance. Further details on the other hyper-parameters setting can be found in the Appendix E.1.

**Constrained clustering.** We test the clustering performance of our model compared with the baselines on four different data sets: MNIST (LeCun et al., 2010), Fashion MNIST (Xiao et al., 2017),

Reuters (Xie et al., 2016) and HHAR (Stisen et al., 2015) (see Appendix A). Note that we pre-processed the Reuters data by computing the tf-idf features on the 2000 most frequent words on a random subset of 10000 documents and by selecting 4 root categories (Xie et al., 2016). Accuracy and Normalized Mutual Information (NMI) are used as evaluation metrics. The results are shown in Table 1. We observe that our model reaches state-of-the-art performance in the Fashion MNIST and Reuters data sets. On MNIST and HHAR data sets, on the other hand, it has comparable clustering performance with C-IDEC, however it is generally more stable accross different data sets.

Table 1: Clustering performances of CVaDE compared with baselines. All methods use $N$ pairwise constraints ($\sqrt{N}$ labels) except the unsupervised VaDE. Means and standard deviations are computed across 10 runs with different random model initialization and pre-training weights.

|  | MNIST | | FASHION | | REUTERS | | HHAR | |
| --- | --- | --- | --- | --- | --- | --- | --- | --- |
|  | Acc | NMI | Acc | NMI | Acc | NMI | Acc | NMI |
| VaDE | 85.9 $_{\pm 5.7}$ | 81.9 $_{\pm 2.2}$ | 59.3 $_{\pm 2.3}$ | 59.8 $_{\pm 1.9}$ | 76.0 $_{\pm 0.7}$ | 50.2 $_{\pm 0.9}$ | 75.3 $_{\pm 5.9}$ | 70.5 $_{\pm 4.0}$ |
| MPCK | 58.8 $_{\pm 0.0}$ | 53.3 $_{\pm 0.1}$ | 57.5 $_{\pm 0.0}$ | 59.8 $_{\pm 1.9}$ | 60.7 $_{\pm 0.5}$ | 50.2 $_{\pm 0.9}$ | 58.1 $_{\pm 0.2}$ | 70.5 $_{\pm 4.0}$ |
| SDEC | 87.9 $_{\pm 0.2}$ | 86.8 $_{\pm 0.3}$ | 59.7 $_{\pm 2.2}$ | 63.0 $_{\pm 1.2}$ | 77.3 $_{\pm 3.9}$ | 60.0 $_{\pm 2.9}$ | 67.0 $_{\pm 7.0}$ | 71.5 $_{\pm 2.0}$ |
| C-IDEC | **97.9** $_{\pm 0.1}$ | **94.3** $_{\pm 0.3}$ | 84.5 $_{\pm 1.7}$ | 76.0 $_{\pm 1.6}$ | 92.6 $_{\pm 4.1}$ | 79.1 $_{\pm 4.1}$ | **93.4** $_{\pm 0.5}$ | **86.3** $_{\pm 1.1}$ |
| **CVaDE** | **98.0** $_{\pm 0.1}$ | **94.4** $_{\pm 0.2}$ | **86.1** $_{\pm 0.2}$ | **77.6** $_{\pm 0.3}$ | **95.3** $_{\pm 0.5}$ | **82.6** $_{\pm 1.4}$ | **93.5** $_{\pm 0.8}$ | **86.5** $_{\pm 1.2}$ |

**Constrained clustering with noisy labels.** In real-world applications it is often the case that the additional information comes from different sources with different noise levels. As an example, pairwise annotations could be obtained by both very experienced domain experts and by less-experienced users. Hence, the ability to integrate constraints with different degrees of certainty into the clustering algorithm is of significant practical importance. In this experiment, we consider the case in which the given pairwise constraints have three different noise levels, $q \in \{0.1, 0.2, 0.3\}$, where $q$ determines the fraction of pairwise constraints with flipped signs (that is, when a must-link is turned into a cannot-link and vice-versa). In Fig. 1 we show the clustering performance of our model compared to the strongest baseline derived from the previous section, C-IDEC. For all data sets, we decrease the value of the pairwise confidence of our method using the heuristic $|W_{i,j}| = \alpha \log \left( \frac{1-q}{q} \right)$ with $\alpha = 3500$. Also, we use grid search to choose the hyper-parameters of C-IDEC for the different noise levels (in particular we set the penalty weight to 0.1, 0.005, and 0.001 respectively). CVaDE clearly achieves better performance in terms of accuracy and NMI on all three noise levels for all data sets. In particular, the higher the noise level the greater the performance difference. We can conclude that our model is more robust than its main competitor. Additionally, C-IDEC cannot model different noise levels within the same data set, while our model can easily include different source of information with different degree of uncertainty.

**Heart Echo.** We evaluate the effectiveness of our model in a real-world application by using infant heart echo cardiogram videos. The data set consists of 305 infant echo cardiogram videos from 65 patient visits obtained from a large University Children's Hospital. Each visit may consist of videos taken from several different angles (called view), denoted by [LA, KAKL, KAPAP, KAAP, 4CV]. We cropped the videos by isolating the cone of the echo cardiogram, we resized them to $64 \times 64$ pixels and split them into individual frames obtaining a total of $N = 20000$ images. We focused on investigating two constrained clustering problems using this data set. First, we cluster the echo video frames by view. This is a relevant task, as in many datasets, heart echo videos are not explicitly labeled (Zhang et al., 2018). Then, we cluster the echo video frames by infant maturity at birth, following the WHO definition of premature birth categories ("Preterm"). We believe that these two clustering tasks demonstrate that our model admits a degree of control in choosing the underlying structure of the learned clusters. As the experiments demonstrate, by providing different pairwise constraints, it is possible to guide the clustering process towards a preferred configuration, depending on what the practitioners are seeking in the data.
For both experiments, we compare the performance of our method, CVaDE, with the unsupervised VaDE method and the C-IDEC. Additionally, we include a variant of both our method and VaDE in

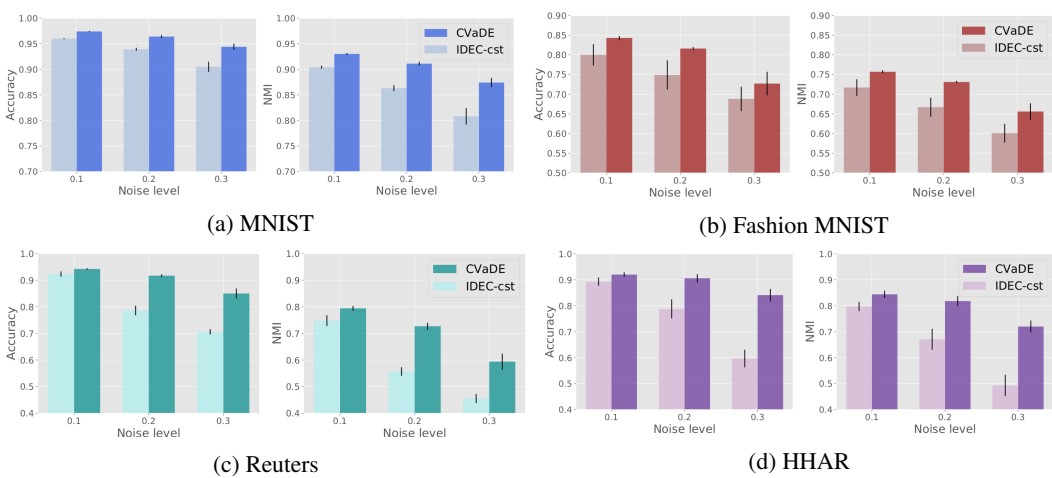

Figure 1: Accuracy and NMI clustering performance on four different datasets with noisy labels.

which we use convolutional layers (CNN-CVaDE and CNN-VaDE), for details on the implementation we refer to the Appendix E.2. The results are shown in Table 2. The CVaDE model outperforms both baselines by a significant margin in both accuracy and NMI in all clustering experiments. This demonstrates that the addition of domain knowledge is particularly effective for medical purposes. Additionally, we observe that C-IDEC performs poorly on real-world noisy data. We believe this is due to the heavy pretraining of the autoencoder, required by DEC-based methods, as it does not always enforce that the learned latent space is suitable for clustering. Additionally, we investigate the relationship between the number of constraints and clustering performance for the view detection task. We observed that with only 5000 pairwise constraints, which could be obtained with less than 80 labels, our model achieves results comparable to those of Table 2 (see Appendix D.3 for details).

Table 2: Constrained clustering results using heart echo cardiogram data with fully connected layers on the left and convolutional layers on the right. Means and standard deviations are computed across 10 runs with different random model initialization and pre-training weights.

| Clustering | Metric | C-IDEC | VaDE | **CVaDE** | CNN-VaDE | **CNN-CVaDE** |
|---|---|---|---|---|---|---|
| View | Acc | $69.5_{\pm 8.4}$ | $33.4_{\pm 2.0}$ | $\mathbf{81.0}_{\pm 2.5}$ | $46.3_{\pm 8.0}$ | $\mathbf{91.4}_{\pm 2.7}$ |
|  | NMI | $47.8_{\pm 15.5}$ | $11.9_{\pm 2.6}$ | $\mathbf{60.9}_{\pm 4.0}$ | $24.2_{\pm 6.5}$ | $\mathbf{80.1}_{\pm 4.7}$ |
| Preterm | Acc | $64.9_{\pm 5.8}$ | $40.9_{\pm 4.6}$ | $\mathbf{73.4}_{\pm 1.3}$ | $40.7_{\pm 4.9}$ | $\mathbf{74.4}_{\pm 2.3}$ |
|  | NMI | $11.0_{\pm 9.9}$ | $6.2_{\pm 1.6}$ | $\mathbf{26.0}_{\pm 2.6}$ | $9.9_{\pm 5.7}$ | $\mathbf{34.7}_{\pm 3.8}$ |

**Face Image Generation** We evaluate the generative capabilities of our model using the UTKFace dataset (Zhang et al., 2017). This dataset contains over 20000 images of male and female faces, aged from 1 to 118 years old, with multiple ethnicities represented. We use a convolutional network for the VAE (the implementation details are described in the Appendix E.3). As a first task, we cluster the data using the gender prior information, in the form of $2N$ pairwise constraints, which requires labels for $0.7\%$ of the data set. Fig. 2a/2b shows the PCA decomposition of the embedding space of both CVaDE and the unsupervised VaDE method. As a second task, we select a sub-sample of individuals between 18 and 50 years of age (approx. 11000 samples), and cluster by ethnicity (White, Black, Indian, Asian) using $2N$ pairwise constraints, Fig. 2c/2d. We believe that one possible reason for biases in machine learning models currently is due to under-representation of various ethnic groups in training data sets. The constrained CVaDE method could be used to identify such imbalances, while requiring only a relatively small number of labeled training samples. This would allow easy identification of such instances of bias in training data sets at a relatively low cost. Furthermore, the ability to generate samples from these detected clusters potentially allows automatic data set augmentation. With the inclusion of domain knowledge, we observe a neat division of the

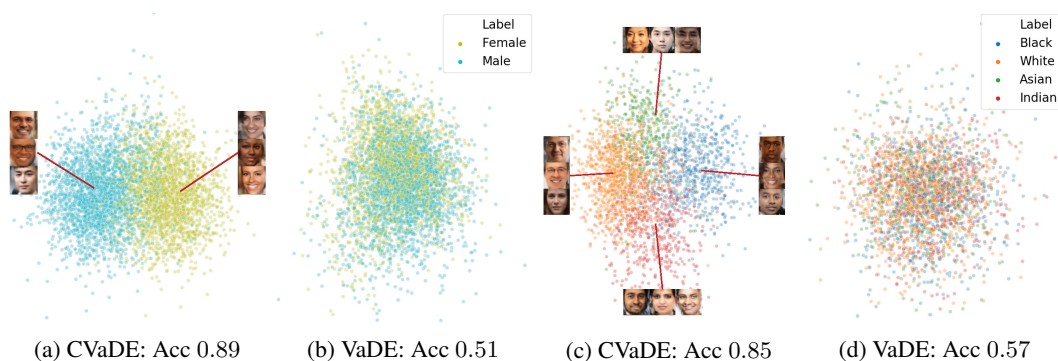

(a) CVaDE: Acc 0.89    (b) VaDE: Acc 0.51    (c) CVaDE: Acc 0.85    (d) VaDE: Acc 0.57

Figure 2: PCA decomposition of test set examples in the embedded space and generative samples using CVaDE and VaDE for (a)-(b) Gender, (c)-(d) Ethnicity. In this configuration, CVaDE obtains a NMI of $0.52$ (gender) and $0.4$ (ethnicity) while VaDE obtains a NMI close to $0$ for both tasks.

selected clusters in the embedding space in both tasks, conversely the unsupervised approach is not able to distinguish any feature of interest.

Finally, using the multivariate Gaussian distributions of each cluster in the learned embedded space, we test the generative capabilities of CVaDE by first recovering the mean face of each cluster, and then generating several more faces from each cluster. Figure 3 shows these generated samples. As can be observed, the ethnicities present in the data set are represented well by the mean face. Furthermore, the sampled faces all correspond to the respective cluster, and have a good amount of variation. The quality of generated samples could be improved by using higher resolution training samples or different CNN architectures.

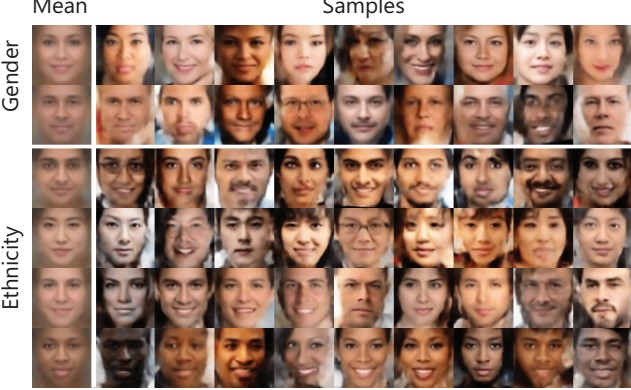

Figure 3: Mean face and sampled faces for each learned cluster, top two rows corresponding to gender, bottom rows to ethnicity

## 5    CONCLUSION

In this work, we present a novel constrained deep clustering method, CVaDE, that incorporates clustering preferences in the form of pairwise constraints, with varying degrees of certainty. In contrast to existing constrained deep clustering approaches, CVaDE uncovers the underlying distribution of the data, resulting in the ability to generate new samples, perform Bayesian model validation and outlier detection. With the integration of domain knowledge, we show that our model can drive the clustering algorithm towards the partitions of the data sought by the practitioners, achieving state-of-the-art constrained clustering performance in real-world and complex data sets. Additionally, our model proves to be robust to noisy constraints as it can efficiently include uncertainty into the clustering preferences. As a result, the proposed model can be applied to a variety of applications where the difficulty of obtaining labelled data prevents the use of fully supervised algorithms.

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

APPENDIX

## A  DATA SETS

The data sets used in the experiments are the followings:

- **MNIST:** It consists of 70000 handwritten digits. The images are centered and of size 28 by 28 pixels. We reshaped each image to a 784- dimensional vector (LeCun et al., 2010).

- **Fashion MNIST:** A data set of Zalando's article images consisting of a training set of 60,000 examples and a test set of 10,000 examples (Xiao et al., 2017).

- **HHAR:** The Heterogeneity Human Activity Recognition (HHAR) dataset contains 10299 sensor records from smart phones and smart watches. All samples are partitioned into 6 categories of human activities and each sample is of 561 dimensions (Stisen et al., 2015).

- **Reuters:** It contains 810000 English news stories (Lewis et al., 2004). Following the work of Xie et al. (2016), we used 4 root categories: corporate/industrial, government/social, markets, and economics as labels and discarded all documents with multiple labels, which results in a 685071-article dataset. We computed tf-idf features on the 2000 most frequent words to represent all articles. A random subset of 10000 documents is then sampled.

- **Newborn echo cardiograms:** The dataset consists of 305 infant echo cardiogram videos from 65 patient visits obtained from a large children hospital. Each visit may consist of videos taken from several different angles, denoted by [LA, KAKL, KAPAP, KAAP, 4CV]. We cropped the videos by isolating the cone of the echo cardiogram, we resized them to 64x64 pixels and split them into individual frames obtaining a total of $N = 20000$ images.

- **UTKFace:** This dataset contains over 20000 images of male and female face of individuals from 1 to 118 years old, with multiple ethnicities represented (Zhang et al., 2017).

## B  CONDITIONAL ELBO DERIVATIONS

In this section, we provide the detailed derivation of the Conditional ELBO with pairwise constraints and we describe how it can be trained efficiently with Monte Carlo sampling. Specifically, the C-ELBO, $\mathcal{L}_{\mathrm{C}}(\mathbf{X}|\boldsymbol{G})$, is the upper bound of the marginal log-likelihood conditioned on $\boldsymbol{G}$:

$$\log p(\mathbf{X}|\boldsymbol{G}) = \log \int_{\mathbf{Z}} \sum_{\mathbf{c}} p(\mathbf{X}, \mathbf{Z}, \mathbf{c}|\boldsymbol{G}) \geq \mathbb{E}_{q_\phi(\mathbf{Z}, \mathbf{c}|\mathbf{X})} \log \frac{p(\mathbf{X}, \mathbf{Z}, \mathbf{c}|\boldsymbol{G})}{q_\phi(\mathbf{Z}, \mathbf{c}|\mathbf{X})} = \mathcal{L}_{\mathrm{C}}(\mathbf{X}|\boldsymbol{G}). \quad (16)$$

By plugging in the marginal log-likelihood of Eq. 3 and the variational distribution of Eq. 6, the C-ELBO can be rewritten as:

$$\mathcal{L}_{\mathrm{C}}(\mathbf{X}|\boldsymbol{G}) = \mathbb{E}_{q_\phi(\mathbf{Z}, \mathbf{c}|\mathbf{X})}[\log p_\theta(\mathbf{X}|\mathbf{Z})p(\mathbf{Z}|\mathbf{c})p(\mathbf{c}|\boldsymbol{G})] - \mathbb{E}_{q_\phi(\mathbf{Z}, \mathbf{c}|\mathbf{X})}[\log q_\phi(\mathbf{Z}, \mathbf{c}|\mathbf{X})] \quad (17)$$

$$\begin{aligned} = {} & \mathbb{E}_{q_\phi(\mathbf{Z}|\mathbf{X})}[\log p_\theta(\mathbf{X}|\mathbf{Z})] + \mathbb{E}_{q_\phi(\mathbf{Z}|\mathbf{X})p(\mathbf{c}|\mathbf{Z})}[\log p(\mathbf{Z}|\mathbf{c})] \\ & + \mathbb{E}_{p(\mathbf{c}|\mathbf{Z})}[\log p(\mathbf{c}|\boldsymbol{G})] - \mathbb{E}_{q_\phi(\mathbf{Z}|\mathbf{X})}[\log q_\phi(\mathbf{Z}|\mathbf{X})] - \mathbb{E}_{p(\mathbf{c}|\mathbf{Z})}[\log p(\mathbf{c}|\mathbf{Z})] \end{aligned} \quad (18)$$

where we used the fact that the variational distribution can be factorized as $q_\phi(\mathbf{Z}, \mathbf{c}|\mathbf{X}) = q_\phi(\mathbf{Z}|\mathbf{X})p(\mathbf{c}|\mathbf{Z})$.

Given that $q_\phi(\mathbf{Z}|\mathbf{X})p(\mathbf{c}|\mathbf{Z}) = \prod_i q_\phi(\mathbf{z}_i|\mathbf{x}_i)p(c_i|\mathbf{z}_i)$ and using Eq 13, we can further factorize Eq 18:

$$\begin{aligned} \mathcal{L}_{\mathrm{C}}(\mathbf{X}|\boldsymbol{G}) = \sum_{i=1}^{N} \Big[ & \mathbb{E}_{q_\phi(\mathbf{z}_i|\mathbf{x}_i)}[\log p_\theta(\mathbf{x}_i|\mathbf{z}_i)] + \mathbb{E}_{q_\phi(\mathbf{z}_i|\mathbf{X})p(\mathbf{c}|\mathbf{z}_i)}[\log p(\mathbf{z}_i|c_i)] \\ & + \mathbb{E}_{p(c_i|\mathbf{z}_i)} \log \pi_{c_i} + \sum_{i,j \neq i} \mathbb{E}_{p(c_i|\mathbf{z}_i)} \mathbb{E}_{p(c_j|\mathbf{z}_j)} \boldsymbol{W}_{i,j} \delta_{c_i c_j} \\ & - \mathbb{E}_{q_\phi(\mathbf{z}_i|\mathbf{x}_i)}[\log q(\mathbf{z}_i|\mathbf{x}_i)] - \mathbb{E}_{p(c_i|\mathbf{z}_i)}[\log q(c_i|\mathbf{z}_i)] \Big] \end{aligned} \quad (19)$$

As $p(c_i|\mathbf{z}_i)$ is discrete, $E_{p(c_i|\mathbf{z}_i)}[\cdot] = \sum_k p(c_i = k|\mathbf{z}_i)[\cdot]$, the above equation can then be written as:

$$
\begin{aligned}
\mathcal{L}_{\mathrm{C}}(\mathbf{X}|\boldsymbol{G}) = \sum_{i=1}^{N} \Big[ & \mathbb{E}_{q_\phi(\mathbf{z}_i|\mathbf{x}_i)}[\log p_\theta(\mathbf{x}_i|\mathbf{z}_i)] + \mathbb{E}_{q_\phi(\mathbf{z}_i|\mathbf{x}_i)}\Big[ \sum_k p(c_i = k|\mathbf{z}_i) \log p(\mathbf{z}_i|c_i = k) \Big] \\
& + \sum_i \sum_k p(c_i = k|\mathbf{z}_i) \log \pi_{c_i} + \sum_{i,j \neq i} \sum_k p(c_i = k|\mathbf{z}_i) p(c_j = k|\mathbf{z}_j) \boldsymbol{W}_{i,j} \\
& - \mathbb{E}_{q_\phi(\mathbf{z}_i|\mathbf{x}_i)}[\log q(\mathbf{z}_i|\mathbf{x}_i)] - \sum_k p(c_i = k|\mathbf{z}_i)[\log p(c_i = k|\mathbf{z}_i)] \Big]
\end{aligned}
\tag{20}
$$

Using the SGVB estimator, we can approximate the above equation as:

$$
\begin{aligned}
\mathcal{L}_{\mathrm{C}}(\mathbf{X}|\boldsymbol{G}) = \sum_{i=1}^{N} \frac{1}{L} \sum_{l=1}^{L} \Big[ & \log p_\theta(\mathbf{x}_i|\mathbf{z}_i^{(l)}) + \sum_k p(c_i = k|\mathbf{z}_i^{(l)}) \log p(\mathbf{z}_i^{(l)}|c_i = k) \\
& + \sum_i \sum_k p(c_i = k|\mathbf{z}_i^{(l)}) \log \pi_{c_i} + \sum_{i,j \neq i} \sum_k p(c_i = k|\mathbf{z}_i) p(c_j = k|\mathbf{z}_j) \boldsymbol{W}_{i,j} \\
& - \log q_\phi(\mathbf{z}_i^{(l)}|\mathbf{x}_i) - \sum_k p(c_i = k|\mathbf{z}_i^{(l)}) \log p(c_i = k|\mathbf{z}_i^{(l)}) \Big],
\end{aligned}
\tag{21}
$$

where $L$ is the number of Monte Carlo samples in the SGVB estimator and it is set to $L = 1$ in all experiments.

## C  VARIATIONAL DEEP EMBEDDING

The Variational Deep Embedding method, VaDE (Jiang et al., 2017), assumes an observed sample $\mathbf{x}_i$ is generated by the following generative process:

$$
c_i \sim Cat(1/K)
$$
$$
\mathbf{z}_i \sim p(\mathbf{z}_i|\mathbf{c}_i) = \mathcal{N}(\mathbf{z}_i|\boldsymbol{\mu}_{c_i}, \boldsymbol{\sigma}_{c_i}^2 \mathbb{I})
\tag{22}
$$
$$
\mathbf{x}_i \sim p(\mathbf{x}_i|\mathbf{z}_i) = \begin{cases} \mathcal{N}(\mathbf{z}|\boldsymbol{\mu}_{x_i}, \boldsymbol{\sigma}_{x_i}^2 \mathbb{I}) & \text{with} \quad [\boldsymbol{\mu}_{x_i}, \boldsymbol{\sigma}_{x_i}^2] = f(\boldsymbol{z}_i; \boldsymbol{\theta}) \text{ if } \mathbf{x}_i \text{ is real-valued} \\ Ber(\boldsymbol{\mu}_{x_i}) & \text{with} \quad \boldsymbol{\mu}_{x_i} = f(\mathbf{z}_i; \boldsymbol{\theta}) \text{ if } \mathbf{x}_i \text{ is binary} \end{cases}
\tag{23}
$$

where $K$ is the predefined number of clusters $\boldsymbol{\mu}_c, \boldsymbol{\sigma}_c^2$ are mean and variance of the Gaussian distribution corresponding to cluster $c$ in the latent space and the function $f(\boldsymbol{z}; \boldsymbol{\theta})$ is a neural network, called *decoder*, parametrized by $\boldsymbol{\theta}$, similarly to a VAE. To infer both the parameters of the Gaussian mixture model and the decoder, the VaDE maximises the likelihood of the data $\mathbf{X}$, that is:

$$
\log p(\mathbf{X}) = \log \int_{\mathbf{Z}} \sum_{\mathbf{c}} p(\mathbf{X}, \mathbf{Z}, \mathbf{c}) \geq \mathbb{E}_{q(\mathbf{Z}, \mathbf{c}|\mathbf{X})} \log \frac{p(\mathbf{X}, \mathbf{Z}, \mathbf{c})}{q(\mathbf{Z}, \mathbf{c}|\mathbf{X})} = \mathcal{L}_{\mathrm{ELBO}}
$$

The variational distribution is chosen to be:

$$
q_\phi(\mathbf{Z}, \mathbf{c}|\mathbf{X}) = \prod_i q_\phi(\mathbf{z}_i, c_i|\mathbf{x}_i) = \prod_i q_\phi(\mathbf{z}_i|\mathbf{x}_i) p(c_i|\mathbf{z}_i) \quad \text{with} \quad p(c_i|\mathbf{z}_i) = \frac{p(\mathbf{z}_i|c_i) p(c_i)}{\sum_k p(\mathbf{z}_i|k) p(k)}
\tag{24}
$$

where $q_\phi(\mathbf{z}_i|\mathbf{x}_i)$ is a Gaussian distribution with mean $\mu(\boldsymbol{x}_i)$ and variance $\sigma^2(\boldsymbol{x}_i)\mathbb{I}$ which are the outputs of a neural network, called *encoder*, parametrized by $\phi$.

The ELBO can be then formulated as:

$$
\mathcal{L}_{\mathrm{ELBO}}(\mathbf{X}) = \mathbb{E}_{q_\phi(\mathbf{Z}|\mathbf{X})} [\log p_\theta(\mathbf{X}|\mathbf{Z})] - D_{KL}(q_\phi(\mathbf{Z}, \mathbf{c}|\mathbf{X}) \| p(\mathbf{Z}, \mathbf{c})).
\tag{25}
$$

## D  FURTHER EXPERIMENTS

### D.1  COMPARISON WITH C-IDEC

In Table 3, we present the quantitative results in term of Adjusted Rand Index (ARI) and the percentage of satisfied constraints (SC) of both our model, CVaDE, and the strongest baseline, C-IDEC with $N$ constraints.

Table 3: Clustering performances of CVaDE compared with C-IDEC. All methods use $N$ pairwise constraints ($\sqrt{N}$ labels) except the unsupervised VaDE. Means and standard deviations are computed across 5 runs with different random model initialization and pre-training weights.

| | MNIST | | FASHION | | REUTERS | | HHAR | |
|---|---|---|---|---|---|---|---|---|
| | ARI | SC | ARI | SC | ARI | SC | ARI | SC |
| C-IDEC | 95.1 $_{\pm 0.5}$ | 100 | 70.5 $_{\pm 2.3}$ | 98 | 88.6 $_{\pm 0.8}$ | 100 | 86.4 $_{\pm 0.8}$ | 100 |
| **CVaDE** | **95.7** $_{\pm 0.2}$ | 100 | **75.5** $_{\pm 0.4}$ | 100 | **90.3** $_{\pm 0.8}$ | 100 | **87.4** $_{\pm 1.0}$ | 100 |

## D.2 NOISY LABELS

In Tables 4/5/6/7, we present the quantitative results in term of Accuracy and Normalized Mutual Information of both our model, CVaDE, and the strongest baseline, C-IDEC with $N$ noisy constraints (presented visually in Fig. 1). In particular, the results are computed for $q \in \{0.1, 0.2, 0.3\}$, where $q$ determines the fraction of pairwise constraints with flipped signs (that is, when a must-link is turned into a cannot-link and vice-versa).

Table 4: Clustering performances with noisy labels averaged over 10 runs on MNIST.

| Noise level | Metric | C-IDEC | **CVaDE** |
|---|---|---|---|
| $q = 0.1$ | Accuracy | $0.960 \pm 0.001$ | $0.974 \pm 0.001$ |
| | NMI | $0.904 \pm 0.003$ | $0.930 \pm 0.002$ |
| $q = 0.2$ | Accuracy | $0.939 \pm 0.003$ | $0.964 \pm 0.003$ |
| | NMI | $0.863 \pm 0.006$ | $0.911 \pm 0.004$ |
| $q = 0.3$ | Accuracy | $0.905 \pm 0.010$ | $0.944 \pm 0.006$ |
| | NMI | $0.808 \pm 0.016$ | $0.874 \pm 0.009$ |

Table 5: Clustering performances with noisy labels averaged over 10 runs on fMNIST.

| Noise level | Metric | C-IDEC | **CVaDE** |
|---|---|---|---|
| $q = 0.1$ | Accuracy | $0.800 \pm 0.027$ | $0.843 \pm 0.005$ |
| | NMI | $0.717 \pm 0.021$ | $0.757 \pm 0.004$ |
| $q = 0.2$ | Accuracy | $0.749 \pm 0.037$ | $0.816 \pm 0.004$ |
| | NMI | $0.667 \pm 0.024$ | $0.731 \pm 0.003$ |
| $q = 0.3$ | Accuracy | $0.688 \pm 0.031$ | $0.727 \pm 0.030$ |
| | NMI | $0.601 \pm 0.024$ | $0.656 \pm 0.021$ |

Table 6: Clustering performances with noisy labels averaged over 10 runs on Reuters.

| Noise level | Metric | C-IDEC | **CVaDE** |
|---|---|---|---|
| $q = 0.1$ | Accuracy | $0.923 \pm 0.010$ | $0.942 \pm 0.003$ |
| | NMI | $0.748 \pm 0.020$ | $0.794 \pm 0.009$ |
| $q = 0.2$ | Accuracy | $0.786 \pm 0.018$ | $0.917 \pm 0.005$ |
| | NMI | $0.557 \pm 0.016$ | $0.727 \pm 0.013$ |
| $q = 0.3$ | Accuracy | $0.706 \pm 0.010$ | $0.850 \pm 0.019$ |
| | NMI | $0.455 \pm 0.017$ | $0.594 \pm 0.030$ |

Table 7: Clustering performances with noisy labels averaged over 10 runs on HHAR.

| Noise level | Metric | C-IDEC | **CVaDE** |
|---|---|---|---|
| $q = 0.1$ | Accuracy | $0.893 \pm 0.016$ | $0.920 \pm 0.009$ |
|  | NMI | $0.797 \pm 0.018$ | $0.844 \pm 0.014$ |
| $q = 0.2$ | Accuracy | $0.788 \pm 0.037$ | $0.906 \pm 0.016$ |
|  | NMI | $0.671 \pm 0.040$ | $0.818 \pm 0.019$ |
| $q = 0.3$ | Accuracy | $0.597 \pm 0.034$ | $0.841 \pm 0.024$ |
|  | NMI | $0.493 \pm 0.041$ | $0.720 \pm 0.023$ |

### D.3  HEART ECHO

**PCA Decompostion**   In Figure 4 we present a PCA decomposition of the embedded space learned by both the CVaDE and VaDE baseline, from the experiment presented in Section 4. It can be observed that the CVaDE model, using N constraints, is able to learn an embedded space which clusters by view much more effectively.

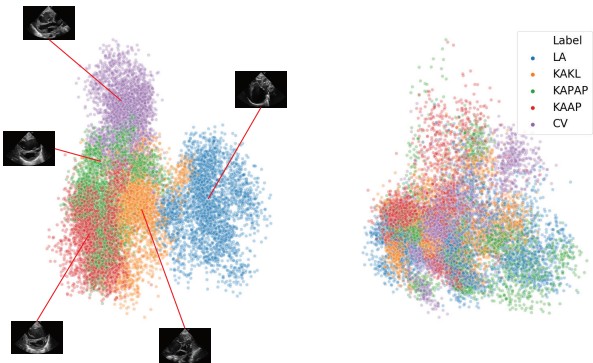

Figure 4: Left: PCA decomposition of CVaDE learned embedded space with representative samples from the dataset, right: baseline VaDE embedded space

**Impact of Constraints**   In order to try to understand how the number of constraints provided impacts the performance of CVaDE, we performed an experiment using the heart echo dataset where progressively more and more constraints were provided to the model during training. Figure 5 demonstrates that the clustering performance of CVaDE improves as more pairwise constraints are provided. This trend continues until 5000 pairwise constraints are provided (requiring $0.35\%$ of the dataset to be labeled), at which point the performance reaches a plateau.

## E  IMPLEMENTATION DETAILS

### E.1  HYPER-PARAMETERS SETTING

In Table 8 we specify the hyper-parameters setting of both our model, CVaDE, and the unsupervised baseline, VaDE. Given the semi-supervised setting, we did not focus in fine-tuning the hyper-parameters but rather we chose standard configurations for all data sets. We observed that our model is robust against changes in the hyper-parameters, except for the batch size. The latter requires a high value, as we simplify the last term of Eq. 13 by allowing the search of pairwise constraints to be performed only inside the considered batch. For the VaDE, we used the same hyper-parameters setting used in their paper (Jiang et al., 2017).

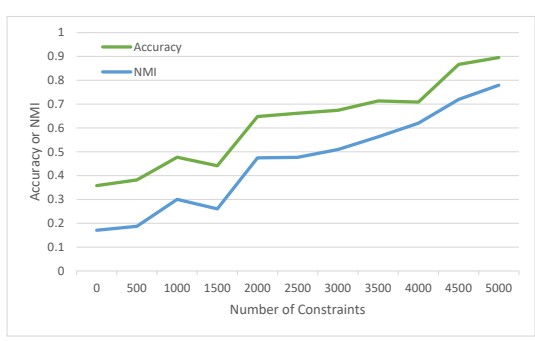

Figure 5: Plot demonstrating that CVaDE's performance improves as more constraints are given. At 5000 pairwise constraints the performance equals that of $N = 20000$ (the total number of samples)

Table 8: Hyperparameters setting of both our model, CVaDE, and the unsupervised VaDE.

|  | CVaDE | | | | VaDE | | | |
|---|---|---|---|---|---|---|---|---|
|  | MNIST | FASHION | REUTERS | HHAR | MNIST | FASHION | REUTERS | HHAR |
| Batch size | 1024 | 1024 | 1024 | 1024 | 128 | 128 | 128 | 128 |
| Epochs | 1500 | 1000 | 1000 | 1000 | 1000 | 1000 | 100 | 500 |
| Learning rate | 0.001 | 0.001 | 0.001 | 0.0001 | 0.002 | 0.002 | 0.002 | 0.002 |
| Decay | $10^{-5}$ | $10^{-4}$ | $10^{-4}$ | $10^{-4}$ | $10, 0.9$ | $10, 0.9$ | $5, 0.5$ | $10, 0.9$ |

### E.2 HEART ECHO

In addition to the model described in Section 4, we also used a VGG-like convolutional neural network. This model is implemented in Tensorflow, using two VGG blocks (using a $3 \times 3$ kernel size) of 32 and 64 filters for the encoder, followed by a single fully-connected layer reducing down to an embedding of dimension 10. The decoder has a symmetric architecture.

The VAE is pretrained for 10 epochs, following which our model is trained for 300 epochs using a learning rate of 0.001 (with an exponential decay of 0.00001), and a batch size of 1024. Refer to the accompanying code for further details.

### E.3 FACE IMAGE GENERATION

The face image generation experiments using the UTK Face dataset described in Section 4 were carried out using VGG-like convolutional neural networks implemented in Tensorflow. In particular,

the input image size of $64 \times 64 \times 3$ allowed two VGG blocks (using a $3 \times 3$ kernel size) of $64$ and $128$ filters for the encoder, followed by a single fully-connected layer reducing down to an embedding of dimension $50$. The decoder has a symmetric architecture.

The VAE is pretrained for $100$ epochs, following which our model is trained for $1000$ epochs using a learning rate of $0.001$ (with a decay of $0.00001$), and a batch size of $1024$. Refer to the accompanying code for further details.

