# OpenReview forum: "A Probabilistic Approach to Constrained Deep Clustering"
_ICLR.cc/2021/Conference — Reject_

### Official Review · AnonReviewer4 · 2020-10-20
**Not good enough**

**Rating:** 4
**Confidence:** 4

**Review:**

This paper solves the constrained clustering from a probabilistic perspective in a deep learning framework. In general, this paper suffers from several major problems. I will illustrate my concerns point-by-point.
1. The authors mention that none of the existing work in the deep (constrained) clustering models the data generative process. First, this is not true. For example, Semi-crowdsourced Clustering with Deep Generative Models. Second, the authors should illustrate the benefits of data generative process for constrained clustering. That is the motivation of this paper. Unfortunately, the motivation is not strong and clear.
2. If I understand correctly, Eq. (9) and (10) are the core techniques for the proposed algorithm. Such a penalty is straightforward in   constrained clustering.
3. In Section 3.4.2, there is another side information, named partition level constraint. The authors might want to explore this as well. This point is not a drawback. Just a suggestion.
4. Some traditional constrained clustering methods with deep VaDE features can be involved for comparisons.
5. It is better to provide some insights on robustness with noisy side information.
6. How to set alpha? Is there some normalization to make alpha within a small range?
7. It is better to show the performance with different numbers of constraints.
8. I am thinking whether two applications in the experimental section are practical in real-world scenarios. I mean how to obtain the pairwise constraints? If I were the project manager in charge of annotation, I will directly label their categories, rather than providing the pairwise constraints.

---

> ### Author Response · Authors · 2020-11-13
> **Response to Reviewer4**
>
> We thank the Reviewer for the feedback, however, we firmly believe that the provided concerns do not justify the given score.
>
> 1. As we clearly state throughout the paper, the advantages of generative models are several: data generation, Bayesian model validation, outlier detection are some of them.  We thank the Reviewer for the additional reference and we will make sure to include it in the related work. The method proposed by Luo et al. 2018 is indeed generative and it shares some similarities with our method. However, they have also several differences, such as the considered graphical model, the prior formulation, and the form of the variational distribution. Additionally, our method shows higher performance in terms of both accuracy and NMI on MNIST when using the same number of constraints (3276), with the accuracy being 0.90 compared to 0.84.
> 2. We argue that Eq. 9 and 10 are not the core techniques of the proposed model, as Eq.1 in the paper of Luo et al. 2018 (Semi-crowdsourced Clustering with Deep Generative Models) is not the core technique of their work. We derived a general framework to include prior information in a variational clustering setting by maximizing the conditional evidence lower bound with the “straightforward” penalty, such derivation is illustrated in the Appendix.
> 3. We appreciate the suggestion.
> 4. We tested the performance of constrained k-means in the latent space of an AE/VAE, it was considerably lower than the rest of the baselines. We believe that training a constrained classical clustering method on VaDE features will not give better performances, as the learned features do not show any structure for more complex datasets (see Fig.2). We will anyway test our hypothesis in future work.
> 5.  It is not clear what it is meant by “provide some insight on robustness”. Fig.1 clearly shows that by increasing the noise level of the given constraints our method outperforms the main competitor. This shows that our model is more robust to noisy information than C-IDEC.
> 6. We tuned the alpha parameter on the MNIST dataset and then we retained the same values for all datasets. We invite the Reviewer to read again “Baseline & implementation details” and “Constrained clustering with noisy labels” of the Experiment section as there are the details on how to set alpha. We investigated different normalizations but we did not include them in the work to retain the simpler prior probability formulation of Eq. 9 and 10.
> 7. We could not include such experiments for lack of space in the main text, but we are happy to include them in a future version of the paper. Our initial results show that our method outperforms the baseline for different numbers of given constraints.
> 8. We argue that it is generally easier to compare than to classify for the non-experts. Additionally, it is always possible to obtain the pairwise constraints from a small subset of labels (as we did for our experiments), whereas the contrary is not possible.

---

> > ### Comment · AnonReviewer4 · 2020-11-14
> > **Response to authors**
> >
> > Thanks for the author response. I agree with some points and would like to increase the scores. But I still remain some concerns.
> > 1. I understand that the generative model can help on the data generation. But how the generative data benefits the clustering task is not verified the experiments. This might be difficult to validate, but some downstream tasks, such as outlier detection can be involved to demonstrate the advantage of generative model.
> > 2. The novelty of the general framework is not rich. This is my major concern I.  In my eyes, the way to incorporate the constraints might be the simplest way, which only has the connection with the indicator assignments.
> > 5. The robustness means that the authors want to generate different noisy labels with the same noise level, different constraints. And the number of constraints is a key factor to understand the proposed algorithm. This is my major concern II. Without this, it is impossible to reimplement the experimental results due to different constraints.
> > 8. Non-experts should not be asked to annotate the label. We actually did some experiments on pairwise labeling. For a dataset with 50 samples, it requests over 1000 pairwise constraints. But it only needs 50 labels for 50 samples. Moreover, such labels can consistently convert to pairwise constraints without any information loss.

---

> > > ### Author Response · Authors · 2020-11-17
> > > **Response to Reviewer4**
> > >
> > > We thank the Reviewer for the additional feedback.
> > >
> > > 1. We believe that learning the distribution of the data is of great advantage in clustering. Being able to generate data from different clusters could prove to be beneficial in many applications. Outlier detection is surely another interesting benefit of generative models and we will include this experiment in a future version of this paper.
> > > 2. We argue that our work is novel as it has not been done before and it successfully guides the clustering toward desirable configurations. Additionally, it is simple and intuitive. Following the Occam's razor principle, we believe those are two great qualities.
> > > 3. We think the Reviewer misunderstood the experiment setting with noisy constraints. The noise level is the fraction of constraints with flipped signs. For each experiment, we sample M pairs of data points and we assign a must-link with probability $1-q$ if they have the same label and a cannot-link with probability $1-q$ if they have different labels. We believe the number of constraints is not necessary a key factor to understand our method, as the generative model and the maximization of the joint probability conditioned on our belief are more important to understand our model than the number of constraints. Finally, it is always possible to consider the same pairwise constraints, however, even with the inclusion of the same constraints the results might vary as it is a semi-supervised method. That is why we averaged the results over 10 runs for both our method and the baselines. Nonetheless, our method is often more stable than baselines.
> > > 4. "*Non-experts should not be asked to annotate the label.* "
> > > We believe that statement to be rather strong and it defies the entire purpose of Citizen science as well as many other crowd-sourced annotations. In general, we would like to remark that many datasets only contain pairwise relations.
> > > "*Such labels can consistently convert to pairwise constraints without any information loss.* "
> > > We do agree and we explained it several times throughout the paper. We believe this is an additional motivation to use pairwise constraints rather than labels for semi-supervised settings.

---

> > > > ### Comment · AnonReviewer4 · 2020-11-17
> > > > **Response to authors**
> > > >
> > > > 3. 10 runs with the same set of constraints or 10 runs with different sets of constraints? I understand the proposed algorithm can tackle different numbers of constraints. But this numbers of constraints will affect the performance. I would like to see whether the performance (average and standard deviation) of the proposed algorithm with different constraints comparing with other methods.
> > > > 4. I am qualified to recognize cat and dog images because human naturally receives visual signals; however, I cannot annotate medical images because I am not an expert in that domain.

---

### Official Review · AnonReviewer2 · 2020-10-29
**An interesting VAE-based method for constrained clustering, however there are a number of concerns with experiments and some claims**

**Rating:** 5
**Confidence:** 5

**Review:**

Summary.

This paper extends the variational deep embedding VaDE model (a VAE-based clustering method) to integrate pairwise constraints between objects, i.e., must-link and cannot-link. The constraints are integrated a priori as a condition. That is, the prior over the cluster labels is conditioned on the constraints. The whole model, referred to as Constrained VaDE (CVaDE), takes the form of a conditional VAE tailored for constrained clustering. Experiments are curried out on various real-world datasets, and the proposed method is compared to VaDE as well as to recent and classical constrained clustering methods.

Strengths.

1. The different ideas used in this paper, such as adopting a mixture of Gaussians as a prior over the VAE latent space for clustering, or the specification of the conditional prior over the cluster labels to integrate pairwise constraints, are not new by themselves. However, combining them together within a VAE framework is interesting and has not been investigated before to my knowledge.

2. The paper is well written, and the proposed method is clearly motivated and described.

3. Experiments are conducted on various data types.

Weaknesses

1. The authors claim superior performance compared to recent constrained deep clustering models. However, looking at the results of Table 1, the proposed CVaDE and the C-IDEC baseline are tight, and the differences in performance do not appear to be statistically significant in most cases.

2. It does not seem like a lot of efforts have been spent for hyperparameters setting.

    a. For instance the same encoder-decoder architecture is used for all datasets, including image and text ones, even though the latter exhibit very different characteristics. In particular, the retained 4-layers architecture (500-500-2000-10) is too complex (very prone to overfitting) for a text dataset such REUTERS, which is extremely sparse, i.e., with very few nonzero entries.

    b. There are important differences in the optimization-hyperparameters (e.g., batch size) used to train CVaDE and its building block VaDE. It would be useful to report the performance of VaDE when trained using the same settings as CVaDE.

3. Despite being a work on constrained clustering, no results regarding the number of satisfied constraints are reported.

4. The authors claim efficiency, but complexity analysis and training time comparisons are missing.

5. Comparisons with baselines when the number of constraints varies are not reported.

6. For the noisy labels experiment, integrating the noise level “q” in the specification of pairwise confidence level is not fair. In practice, we may not always have access to such information in a context of unsupervised learning.

Additional comments and questions.

1. For tractability purposes, the cluster proportions are all set to be equal (1/K). It would be useful to investigate the impact of this assumption on datasets exhibiting very unbalanced cluster sizes. One possibility is to preprocess some of the considered datasets to create such case.

2. Performance is assessed using Normalized Mutual Information (NMI) and Accuracy.  I would suggest reporting the Adjusted Rand Index (ARI) as well. The latter metric is particularly suitable in a context of constrained clustering, as it measures the proportion of pairs of objects clustered similarly according to both the predicted and the ground truth partitions.

3. Have you considered conditioning the variational posterior on the constraint information G?

4. Are you using a held-out test set for evaluation?

---

> ### Author Response · Authors · 2020-11-13
> **Response to Reviewer2**
>
> We thank the Reviewer for the feedback. We provide below a detailed response to each point raised by the Reviewer.
>
> Weaknesses:
>
> 1. We argue that the differences in performance are statistically significant in half of the standard datasets (Table 1), as we mention in the paper.  Additionally, our method outperforms the main competitor by a large margin if noisy labels are considered (which represents the real world scenario) and if a more complex dataset is used (echos).
> 2.  As written by Reviewer #1:  “ *Clustering being unsupervised (here semi-supervised) one should not (rather cannot) employ different hyper-parameters for a different dataset.* ”. We would like to point out that we did not fine-tune the hyper-parameters for different datasets. (a) The same network architecture has been used for the standard datasets for the simple reason of performing a fair comparison with the baselines. (b) Throughout our experiments VaDE has shown to be quite sensitive to different hyper-parameters settings, its performance is very low when trained with the same setting as CVaDE.
> 3. We appreciate the suggestion and we will also include the percentage of satisfied constraints in a revised version of the paper.
> 4. The complexity and training time equals those of the VaDE method. We will make sure to include this in the paper.
> 5. We could not include such experiments for the lack of space in the main text, but we are happy to include them in a revised version of the paper.
> 6. We argue that, in a real-world scenario, one can estimate the noise level of the given constraints, depending on the level of expertise of the sources. Nonetheless, our method is quite robust also with different choices of alpha, hence the noise level does not have to be exact. We would like to point out that the hyper-parameters of the baseline must be fine-tuned for each level of noise using the labels. On the contrary, our method does not need fine-tuning. The simple heuristic we derived on the MNIST dataset work for all datasets.
>
> Additional comments and questions:
>
> 1. This is an interesting point and it has not been discussed by both VaDE and GMM-VAE (which also uses the hypothesis of balanced data). We will investigate it in a future version of the paper.
> 2. We thank the Reviewer for the suggestion and we will include ARI in a revised version of the paper.
> 3. Yes, we considered it. However, as we state in Section 3.2, we choose to retain instead a mean-field variational distribution. If conditioned on G, the latter does not hold. This is because the cluster assignments, conditioned on the pairwise prior information, are not independent.
> 4. Yes

---

> > ### Comment · AnonReviewer2 · 2020-11-19
> > **Response to authors**
> >
> > Thanks for the response. The authors answered my questions, and they promise to address some of my concerns in a future version of the paper. That said there are still some remaining issues (please see my response to the rebuttal), and based on the current content of the paper my original assessment remains unchanged for now.
> >
> > R1. Significance of the results, noisy data in real-world scenarios, echo is more complex.
> >
> >     a. On the significance of the results on two datasets (Table 1). This should be supported by statistical test results, and the claims (in the abstract, introduction and conclusion) regarding the superiority of CVaDE compared to recent constrained deep clustering models should be adjusted accordingly.
> >
> >     b. Even in real-world scenarios, the rate of noisy labels may be very low. And I agree with reviewer #4 on the fact that it does not make sense to ask non-experts to label data or specify constraints.
> >
> >     c. How is Ehco more complex than the other datasets?
> >
> > R2. Hyperparameter search in unsupervised scenarios and performance of VaDE.
> >
> >     a. Under the unsupervised setting, as long as you do not make use of the ground truth labels, there are different ways to choose hyper-parameters. One possibility is to rely on information criteria such as BIC, or simply leverage some prior knowledge. For instance as mentioned in my original comment, the retained neural architecture is too complex for an extremely sparse dataset such as REUTERS. I would recommend looking at this paper [1, section 3.1], which uses a much simpler architecture for text data (as compared to image data) owing to sparsity.
> >
> >     b. Regarding VaDE, the original paper reports much higher clustering accuracy on the same datasets (e.g., MNIST). In the appendix E.1 it is mentioned that the same hyperparameter settings as in the original paper are used. How would you explain such discrepancy? Is it due to differences in other experimental settings?
> >
> > R3.	The complexity and training time equals those of the VaDE method.
> >
> >     a. Compared to VaDE, the integration of the constraint information in CVaDE would introduce an additional computational overhead. So by equal training time here do you mean comparable?
> >
> > R4.	Estimation of noise level, hyperparameter-search for C-IDEC.
> >
> >     a. It is still not clear how to estimate the noise level under a fully unsupervised scenario. How would one measure the level of expertise? Can you provide a concreate example (maybe with the Echo dataset)?
> >
> >     b. Since CVaDE is not very sensitive to the value of alpha, it would be interesting to investigate how it performs under different noise levels without decreasing the value of the pairwise confidence. This would provide evidence on the importance of decreasing the confidence level.
> >
> >     c. How is the hyper-parameter search performed for C-IDEC? In particular, what is the search space for the penalty weight? Looking at the results of Figure 1, it seems like C-IDEC is even worse than VaDE (unconstrained) when the noise level is 0.3. This suggests that the penalty weight for the constrain-loss in C-IDEC is maybe high. One easy way to verify this, is to check the performance of C-IDEC when the penalty weight is set to zero corresponding to the unconstrained case.
> >
> > R5. Using held-out test set.
> >
> >     a. The information regarding the train/test-splitting are missing in the paper, and it would be useful to include them.
> >
> >     b. Moreover, in the accompanying code, it seems like pretraining is performed on both the train and test sets (pretrain function the main.py file). Can you please clarify this aspect?
> >
> > [1] Van Der Maaten, Laurens. "Learning a parametric embedding by preserving local structure." Artificial Intelligence and Statistics. 2009.

---

> > > ### Author Response · Authors · 2020-11-23
> > > **Response to Reviewer2**
> > >
> > > We thank the Reviewer for the additional feedback and we integrated some of the experiments suggested by the Reviewer (see Appendix D.1). Further experiments will be included in the camera-ready version of the paper or for a future re-submission.
> > >
> > > R1. We would like to point out that the results in bold are statistically significant as we tested it with a dependent t-test for paired samples, obtaining a statistic around 5.57 and a p-value of 0.0008 for fMNIST and we obtained similar results for Reuters. We also argue that the claims in the introduction/abstract are TRUE as our method shows superior clustering performance for a majority of datasets (hence a wide range of datasets) and it is more robust than the competitors. Echocardiograms are considered a challenging dataset [1] for several reasons: 1. the presence of speckle noise in ultrasound images frequently limits their contrast, affecting both human interpretation and computer-assisted analysis, 2. the angle at which they have been recorded is not stable and it may often vary among patients, 3. different machines may also be used to record echos producing a bias in the dataset.
> > >
> > > R2. (a) We thank the Reviewer for the suggestion. Unfortunately, comparing clustering results using baselines with different architectures does not result in a fair comparison, as different networks produce different results. We will try changing the architecture of all baselines to reduce the complexity in a future resubmission. (b) It is important to note that VaDE reports the highest results obtained over a wide range of runs while we report the mean and the standard deviation. As VaDE is an unsupervised method, the results greatly vary among different runs.
> > >
> > > R3. We apologize for the ambiguity of our answer. The training time of CVaDE is comparable to VaDE's. However, the computational overhead is O(L^2), where L is the batch size.
> > >
> > > R4. (a) In the medical setting, the noise level can be guessed from the level of expertise of the sources (doctors for example) and it can be adjusted by asking different sources to label the same subset of the dataset and comparing the differences to the most reliable source. (b) Without decreasing the pairwise confidence level we expect lower performance, however, preliminary results show that, even in this setting, CVaDE would still be superior to C-IDEC. We will include further investigation in the camera ready version/future re-submission. (c) The search space is from 0 to 0.1 (the value used for true labels). It is worth noting that the unsupervised IDEC method has lower performance than the VaDE method. Indeed, with 0.3 noise level the performance of IDEC and C-IDEC are quite similar.
> > >
> > > R5. The pre-training does not use any label information as it is unsupervised, for this reason, we use the entire dataset. As an example, VaDE does not use the train/test split for evaluating its performance as it is completely unsupervised.
> > >
> > > [1] Leclerc, S. et al. “Deep Learning for Segmentation Using an Open Large-Scale Dataset in 2D Echocardiography.” IEEE Transactions on Medical Imaging 38 (2019): 2198-2210.

---

### Official Review · AnonReviewer1 · 2020-10-31
**Deep Constrained Clustering**

**Rating:** 5
**Confidence:** 5

**Review:**

**Summary**
This work proposes CVaDE which is an extension of variational based deep clustering model (VaDE) with additional incorporation of prior clustering preferences as supervision. These priors guide the underlying clustering process towards a user-desirable partitioning of input data. The priors are provided in the form of pairwise constraints indicating which pair of samples belongs to same or different class. Clustering process is modelled using variational Bayes in which the clustering constraints are incorporated into prior probabilities with varying degree of uncertainty. The empirical results shows that in comparison to unconstrained clustering the small amount of pairwise constraints significantly improves clustering performance. Further, it demonstrates CVaDE's robustness to noise, generation capability as well as successful incorporation of different desirable preferences to drive clustering performance towards completely different partitioning.

**Quality**
The paper is well written albeit with numerous typographical error (some of which are listed at the end of this review). Experimental evaluation seems thorough. However, I would like to compare results on complex datasets as well as with large classes (> 10). Complex datasets includes STL-10, YouTube Faces, mini ImageNet etc. Please show efficacy on diverse sets of data covering large variation in number of classes, dimensionality, attributes.

Moreover, clustering being unsupervised (here semi-supervised) one should not (rather cannot) employ different hyper-parameters for different dataset. Under the context of zero ground truth data availability, they should rather be fixed. Table 7 says otherwise.

**Originality**
As mentioned above, CVaDE is extended from VaDE but with prior input constraints. Thus the conditional ELBO loss objective is thus a simple extension of VaDE objective. Apart from this, the prior distribution used for pairwise constraints is adapted from work of Lu & Leen (2004). In summary, the work carries very little novelty.

**Significance**
Constrained clustering has been around for some time in various forms. However, the subtle difference CVaDE brings to the table is how to incorporate them into prior probabilities.

Like VaDE, CVaDE is also clustering cum generative model. Once trained, model can be employed for sampling new data. Due to better training procedure using constraints, the generated samples is bound to be perceptually better. However, the samples are not better than the state of the art conditional generative models such as InfoGANs.

**Clarity**
1. In eq(2), shouldn't it be $\mu_{z_i}$ instead of $\mu_{x_i}$. Is function $f(z_i; \theta)$ not deterministic ? My understanding is given fixed $\mu_{z_i}$ one can sample as many $x_i$. Same goes for $\sigma_{x_i}$.
2. Figure 5, axis labels are missing.
3. Under experiments, please make clear what are we solving for - $z$ and $c$ ? Have you tried k-means on extracted $z$ post training ?
4. What is penalty weight ? I did not find any description.
5. Why C-IDEC cannot model different noise levels within same data set ?
6. Where is the derivation for Conditional ELBO formulation ? In appendix I only find solution to C-ELBO not how to derive Eq (5).
7. What is the impact of imbalanced dataset on CVaDE ? I presume apriori this imbalance is not known to the user.
8. Eq (19), is $\mathbb{E}$ different from $E$ ?
9. Eq (19), Eq (20) summation w.r.t. is pulled out. Typo in $W_{ij}$ component.
10. Eq (21), some of the terms are approximated by Monte carlo sampling while others are still taking expectation
11. In Eq (18), If 3rd term is marginalised w.r.t. $q(Z|X)$ then it is technically wrong to apply monte-carlo sample to central line in Eq (21). Remember $\frac{1}{L}$ approximates $q(z_i|x_i)$ which is applicable for 1st, 2nd and 4th terms. Not for all.
12. Eq(12) $\delta_{c_ic_j}$ is missing

---

> ### Author Response · Authors · 2020-11-13
> **Response to Reviewer1**
>
> We thank the Reviewer for the detailed feedback. We apologize for some typographical errors present in the Appendix, we improved it in the revised version of the paper. We provide below a detailed response to each point raised by the Reviewer.
>
> *I would like to compare results on complex datasets as well as with large classes (> 10). Please show efficacy on diverse sets of data covering large variation in the number of classes, dimensionality, attributes.*
>
> We argue that we have already considered two quite complex datasets (noisy echos from newborns and the UTKFace dataset). These two datasets, together with the more standard ones, show great variability in terms of dimensionality, attributes, and the number of classes (2, 5, 10, 6).  We will consider including a dataset with more classes in a future version of this paper.
>
> *In summary, the work carries very little novelty.*
>
> Novelty is a very discussed topic and, unfortunately, a very subjective one. We think our model considers a different approach compared with previous work in constrained/semi-supervised clustering. Moreover, we derive a rather general framework for incorporating prior knowledge in variational deep clustering, which is missing in the literature.
>
> *However, the samples are not better than the state of the art conditional generative models such as InfoGANs.*
>
> InfoGAN surely has many advantages, among them the reconstruction quality is one of them. However, it does not perform clustering and, therefore, should not be considered as a comparison. Additionally, the quality of the reconstruction can be easily improved by using more complex network architectures.
>
> *Clarity*
>
> 1.  No, $\mu_z, \sigma_z$ are the mean and variance of the gaussian in the latent space while $\mu_x, \sigma_x$ are the mean and variance of the gaussian in the input space. Function $f(z_i;\theta)$ is deterministic.
> 2. We fix Fig. 5 in the revised version of the paper.
> 3. We already performed some experiments on constrained k-means in the latent space of an AE/VAE and the performances were considerably lower than the rest of the baselines. We did not try to perform k-means on the extracted $z$, as our method already learns the parameters of a mixture of Gaussians in the latent space which is more informative than K-means.
> 4. (5.) The penalty term is described in Zhang et al, 2019b (C-IDEC) and it is the weight associated with the pairwise loss in their objective. The authors do not provide any guidance on how to choose it and it does not vary between data points.
> 6. The C-ELBO can be easily derived and we have included it in the revised version of the paper (Appendix B).
> 7. This is an interesting point and it has not been discussed by both VaDE and GMM-VAE (which also use the hypothesis of balanced data). We will investigate it in a future version of the paper.
> 8. (9-10-11-12) We have corrected the typos in the revised version of the paper.

---

### Decision · Program_Chairs · 2021-01-07
**Final Decision**

**Decision:**

Reject

**Comment:**

We thank the authors for their detailed responses to reviewers, and for engaging in a constructive discussions.

As explained by the reviewers, the paper is clearly written and the method is novel. However, the novelty is to combine existing ideas and techniques to define an objective function that allows to incorporate cluster assignment constraints, which was considered incremental. Regarding quality, the discussion highlighted some possible improvements that the authors propose to do in a future version of the paper, and we encourage them to follow that direction. Regarding significance, although the experimental results are promising there were some concerns that the improvement over existing techniques is marginal, and that more experiments leading to a clearer message would be useful.

In summary, this is not a bad paper, but it is below the standards of ICLR in its current form.